# Optimization of Three Key Geometries of a Steam Ejector under Varied Primary Nozzle Geometries

**DOI:** 10.3390/e25010015

**Published:** 2022-12-21

**Authors:** Jia Yan, Ruixin Li, Chen Wang

**Affiliations:** 1School of Civil Engineering and Architecture, Southwest University of Science and Technology, Mianyang 621010, China; 2School of Control Science and Engineering, Shandong University, Jinan 250061, China

**Keywords:** steam ejector, primary nozzle, CFD simulation, entrainment ratio, converging and diverging section

## Abstract

In this paper, three key geometrical dimensions such as XL3 (constant pressure mixing chamber length), D5 (diameter of the throat of the ejector), and XL5 (length of the throat of the ejector) were separately or sequentially optimized under different lengths and angles of two sections of the primary nozzle. Furthermore, shock cluster number and shock chain length or area of low static pressure zone were used to analyze the effect of geometries on ejector performance, which is normally indicated by entrainment ratio (ER, or the ratio between the mass flow rate of secondary flow and the mass flow rate of primary flow). The results show that: (1) the improvement of ejector performance with only optimization of the primary nozzle is quite limited, in comparison, the impact of length and angle of nozzle diverging section on ejector performance is higher than that of converging section; (2) the relative sensitivity of ER to three key dimensions is much higher than that to the angles and lengths of the primary nozzle; (3) with the increase of XL3 and D5, ER needs a larger angle and smaller length of nozzle converging section; (4) the impact of key geometries on ejector performance can be analyzed with the help of shock cluster number and shock chain length or area of low static pressure zone.

## 1. Introduction

Carbon capture used in flue gas emission is one of the most important methods of reducing CO_2_ [1]. However, the absorption and resolution of CO_2_ in the process of carbon capture consume a big amount of thermal output of the plant. As a component with simple structure and easy maintenance, the ejector is widely used in hydrogen energy [2], refrigeration, and chemical industry [3,4]. Zhang et al. [5] applied an ejector to collect CO_2_ from flue gas, which is greatly obvious in reducing energy consumption in the traditional carbon capture process. Moreover, three ways to drive the ejector were used to evaluate the potential energy-saving effect [6].

In this condition, the ejector performance which is based on the optimization of the geometries directly affects the energy-saving effect in the carbon capture system. Sriveerakul et al. [7,8] optimized the ejector primary nozzle with this application, which made it convenient for subsequent researchers to optimize the primary nozzle through CFD simulation. Sun et al. [9] studied the converging and diverging sections of the primary nozzle through CFD simulation, and the entrainment ratio (ER) of the ejector increased by 19.79%. Sheng et al. [10] proposed an improved alternating-lobe nozzle, studied the fluid state of the alternating-lobe nozzle, and claimed that the new nozzle can promote high fluid mixing efficiency in the core area. In addition, Li et al. [11] studied the steam expansion characteristic at the primary nozzle. Yan et al. [12] optimized ejector primary nozzles, and different optimal angles and lengths were in the converging and diverging sections of the primary nozzle. Fu et al. [13] optimized a steam ejector, and the nozzle throat diameter ranging from 4.8 to 4.85 can increase ER by 25%. Thongtip et al. [14] also investigated the throat diameter of the nozzle. Their study showed that different fluid expansion coefficients happened at the nozzle outlet. It was suggested that the throat diameter should be determined according to the actual situation. Fu et al. [15] recommended that the different outlet diameter of the primary nozzle has a certain effect on ER.

In addition, the optimization of other key geometries of the ejector such as nozzle exit position (NXP), and lengths of both constant-pressure and constant-area mixing chambers was investigated by some scholars. Chen et al. [16] identified the relationship between NXP and ER, and results showed that the critical ER is determined by the secondary flow choking. Zohbi et al. [17] found that backflow happened to the ejector when NXP was set at a certain value. Furthermore, Zhu et al. [18] optimized the NXP under different constant area mixing chambers. Meanwhile, Dong et al. [19] optimized the constant area mixing chamber length. Varga [20] optimized the diameter of the constant-area mixing chamber and NXP. Similarly, Reis et al. [21] evaluated the effect of mixing chamber length. Furthermore, our previous research [22] was involved to optimize three key geometries of XL3, XL5, and D5 under various optimization sequences.

The ejector consists of five parts: nozzle, secondary flow induction chamber, two mixing chambers, and diffuser. The three key geometries largely affect ejector performance [22]. So far, however, the effect of the primary nozzle structure on the optimum of three key geometries of a steam ejector (Figure 1) has not been mentioned. Hence, the following works will be carried out in this study:The angle and length of both converging and diverging sections of the primary nozzle are optimized, respectively.With the given angle and length of the converging section of the primary nozzle, three key geometries were optimized separately or with the sequence of XL3→D5→XL5. Thus, the optimum of the three key dimensions and the corresponding maximum ER are identified.With the given angle and length of the diverging section of the primary nozzle, three key geometries were optimized separately or with the sequence of XL3→D5→XL5. The impact of the angle and length of the diverging section of the primary nozzle on the optimal values of the three key dimensions and relevant maximum ER is determined.Relative sensitivity of ejector performance to primary nozzle geometries and the other three key geometries is evaluated.The effect of the ejector geometries on ejector performance is analyzed with shock cluster number and shock chain length or area of low static pressure zone inside the ejector.

## 2. Experimental Rig

The working principle of the ejector-based carbon capture system can be referred to in Ref. [23], and the initial geometries of the ejector are given in Table 1.

Figure 2 shows the schematic of the ejector-based test bench. The sensors used are as follows: “T” for thermometers; “P” for manometers; and “GF” and “FT” for vortex flowmeters and orifice flowmeters.

## 3. CFD Modeling and Model Validation

The derivation of conservation equations of energy, mass, and flow related to the calculation is presented in our previous work [22]. A realizable k-ε turbulence model with better performance prediction [24,25] is selected in the simulation.

Figure 3 is a structural grid created by using the software Gambit [26]. Fluent 19.0 is used to carry out the CFD simulation. Implicit double precision and second-order upwind discrete schemes are utilized [27]. The energy equation residual is set to less than 10^−6^, while the rest residuals are less than 10^−5^. The pressure relaxation factor is set to 0.1 and all others are set to 0.3. Boundary conditions are illustrated in Table 2. 

Under the grid numbers 119,720, 161,600, and 23,730, the differences in axial static pressure are all within 1% as shown in Figure 4. Considering both the calculation accuracy and time, thus, the medium one with 161,600 is adopted for the simulation.

Figure 5 is the CFD and experimental ER, in which, ER is defined as:(1)$$\mathrm{ER}=\frac{{\dot{M}}_{s}}{{\dot{M}}_{p}}$$
where ${\dot{M}}_{p}$ and ${\dot{M}}_{s}$ are the primary and secondary flow mass flow rates, respectively. The average and maximum discrepancies of ER are 10.3% and 13.8%, respectively. The CFD results are quite close to experimental results, so the CFD model can be used for this study.

## 4. Results and Discussion

### 4.1. Effect of the Converging Section of the Primary Nozzle

(1)Ejector performance under varied lengths and angles of converging section

Figure 6 is the impact of L_1_ on ER when other dimensions remain constant. When the L_1_ increases from 30% (19.5 mm) to 110% (71.5 mm), the highest ER is 0.318 (the length of the converging section is 80%), thus the optimal ER increases by 2.9% over the initial ER.

Figure 7 displays the impact of θ_1_ on ER. When θ_1_ increases from 60% (4.63°) to 140% (10.81°), ER reaches the maximum of 0.317 (the corresponding θ_1_ is 90%), thus, the ER increases by 2.4% over the initial ER.

In order to compare the relative sensitivity of ER to L_1_ and θ_1_, the following equation is introduced to calculate the relative sensitivity: (2)$$S=\frac{\frac{ER_{m}-ER_{r}}{ER_{r}}}{\frac{\left|G_{m}-G_{r\;}\right|}{G_{r}}}$$
where, *S* is the relative sensitivity of ER; *ER_m_* and *ER_r_* are the maximum ER and the ER under the referred percentage of the geometry; *G_m_* is the percentage of the geometry under maximum ER; and *G_r_* is the referred percentage of the geometry.

For varied L_1_, the maximum ER is 0.318 obtained at 80% of the initial L_1_, and *ER_r_* and *G_r_* are 0.31 and 100% of the initial L_1_, respectively, thus, the S to L_1_ is 0.1032. The S to θ_1_ is 0.2455. In comparison, the S to θ_1_ is relatively higher than that to L_1_. However, optimizing both L_1_ and θ_1_ has quite little performance improvement for the ejector, hence, it is necessary to further optimize other key geometries with varied L_1_ and θ_1_. As a result, the aim of the next section is to optimize the three key geometries under varied L_1_ and θ_1_, to evaluate the impact of varied L_1_ and θ_1_ on the optimized three key dimensions as well as the corresponding maximum ER. Especially, the three key dimensions are individually optimized with the varied L_1_ and θ_1_.

(2)Individually optimizing each of the other three key geometries under varied lengths and angles of the nozzle converging section

Figure 8 shows the obtained ER when XL3 is optimized under a varied percentage of L_1_. The change of ER with the percentage of initial XL3 under the varied percentage of L_1_ is presented in Figure 8. When XL3 is 20% of the initial value, the ER under L_1_ of 78 mm (120% of the initial value) is the maximum. When XL3 is increased to 30% of the initial value, the maximum ER can be obtained when L_1_ is 78 mm (120%). However, when XL3 is increased to 40% of the original size, there is almost no difference in ER at different L_1_ and the maximum ER is achieved at L_1_ of 52 mm (80%). When XL3 is increased to 50%, the highest ER can be reached when L_1_ is 52 mm (80%). Moreover, when XL3 is increased to 60%, the highest ER is obtained at 65 mm of L_1_ (100%). Obviously, a shorter converging section of the primary nozzle is needed to achieve higher ER as the increase of XL3. In addition, for given conditions, the highest S to L_1_ and XL3 are 0.146 and 0.243, respectively; thus, the S to XL3 is much larger than that to L_1_.

It is attempted to use the Mach number contour as illustrated in Figure 9 to analyze the impact of XL3 when XL3 increases from 20% (4 mm) to 60% (12 mm) with the interval of 20% when the L_1_ is 100% of its initial value. For the given three cases of XL3, the number of shock clusters in the ejector is four, and the shock intensity and the shock chain length are quite close to each other. Therefore, it is almost impossible to distinguish the difference between XL3 on ER from shock cluster number, intensity, and shock chain length. Hence, the static pressure (30–130 kPa) distribution contour is considered to analyze the effect as displayed in Figure 10. When XL3 is 20% of the initial value, the static pressure of the second and third shock clusters is the lowest, the distribution area of the low static pressure is relatively large, and the low static pressure area is quite close to the suction chamber, which has relatively strong entrainment ability. When XL3 is increased to 40% of the initial value, the low static pressure zone is bigger than in the former case, and the distance between the low static pressure zone and the suction chamber is shorter. Therefore, the ejector performance is further improved and ER reaches the maximum value. Then, when XL3 increases to 60% of the initial value, the area of the low static pressure zone become smaller, and the area of the high static pressure zone between the adjacent low static pressure zones increases, which has a negative entrainment impact. 

When both XL3 and XL5 are the initial values, the ER with D5 in varied L_1_ is shown in Figure 11. When L_1_ increases from 40% (26 mm) to 100% (65 mm), the largest ER is achieved at 94% of the initial D5. However, when L_1_ continues to increase to 120%, the D5 corresponding to the largest ER decreases to 93%. Furthermore, for all the given D5, the maximum ER can be obtained at 80% (52 mm) of the initial L_1_. With the same relative sensitivity analysis method, the maximum S to L_1_ is 0.203 and that to D5 is 3.63. It can be seen that S to D5 is much greater than that to L_1_.

When both the XL3 and D5 are the initial values, the ER with XL5 at varied L_1_ is presented in Figure 12. When L_1_ varies from 40% to 100%, the highest ER is obtained at 50% of the initial XL5. When L_1_ rises to 120%, the best XL5 is achieved at 70% of the initial value. Moreover, for given XL5, the calculated largest S to L_1_ under different XL5 is 0.409, while the maximum S to XL5 under different L_1_ is 0.424, thus, the difference between the two can be ignored. 

Therefore, according to the above-mentioned results, it can be seen that the varied L_1_ has a different impact on the optimal three key dimensions and associated maximum ER. The impact of varied θ_1_ on the optimal three key dimensions and associated maximum ER is investigated as follows. 

Figure 13 shows the obtained ER when XL3 is individually optimized under different θ_1_. When XL3 increases from 30% to 50% of the initial size at an interval of 5%, ER gradually increases first and then decreases. When XL3 varies from 30% to 40% of the initial size, ER reaches the maximum value at θ_1_ of 60% of the initial value; when XL3 achieves 45% of its initial value, the θ_1_ is 80% of the initial value (6.617°) which offers the maximum ER; and when XL3 achieves 50% of its initial value, the maximum ER can be gained when the θ_1_ is 100% of the initial value. Thus, with the increase of XL3, larger θ_1_ is needed to achieve higher ER.

In the same way, when D5 is optimized alone under varied θ_1_, the optimization results are presented in Figure 14. The optimal D5 is at 94% of the initial value except for that of 80% of the initial θ_1_. In addition, the maximum ER is obtained when θ_1_ is the original value. In addition, with the increase of D5, larger θ_1_ is required to achieve better ER.

Likewise, the variation of ER with XL5 under varied θ_1_ is illustrated in Figure 15, when both the XL3 and D5 are the initial values. When θ_1_ is 60% of the initial value, the optimal XL5 is obtained at 65% of the initial value; when θ_1_ increases to 80%, 100%, and 120% of the initial value, the optimal XL5 increases to 70%; when θ_1_ continues to increase to 140%, the optimum XL5 increases to 75%. That is to say, with the increase of XL5, larger θ_1_ is required to reach a larger ER.

From the above optimization results, for most of the cases, with the increase of the three geometries, larger L_1_ and θ_1_ are required to achieve higher ER; however, the ejector performance improves limited with the three geometries optimized individually. As mentioned in our previous study [23], the ER can be obviously enhanced if the three geometries are optimized in the order of XL3→D5→XL5, therefore, the key geometries under different L_1_ and θ_1_ are optimized in this order as presented in the following section.

(3)Sequentially optimizing the three key geometries under varied lengths and angles of the nozzle converging section

With the optimized XL3, the optimization of D5 under varied length of the nozzle converging section is shown in Figure 16.

As presented in Figure 16 and Table 3, the ER with D5 at varied L_1_ is basically similar. As D5 increases from 98% of the initial value to 100%, there is a negligible difference between ER with L_1_ for each D5. When D5 increases to 101% of the initial size, the gap of ER at different L_1_ is relatively evident, and the maximum ER achieves at L_1_ of 39 mm (60%). However, when D5 increases to 102% of the initial size, ER reaches its maximum value at L_1_ of 65 mm (100%). Thus, with the increase of D5, a larger L_1_ is needed to achieve higher ER. The above-mentioned results are similar to that of Figure 11; however, the difference between them is that the optimal D5 and corresponding maximum ER under the sequential optimization are larger than those under individual optimization.

Under the same sequential optimization of XL3→D5→XL5, the f ER with XL5 at varied L_1_ is displayed in Figure 17. For each XL5, ER achieves the maximum value when L_1_ is 52 mm (80%). 

The change of ER with D5 based on optimal XL3 under different θ_1_ is presented in Figure 18. As shown in Figure 18, for D5, the maximum ER is obtained at 100% of the initial D5 under given θ_1_. However, when D5 rises from 98% to 99% of the original size, the ER under the θ_1_ of 4.26° is greater than that of other cases of θ_1_; when D5 is increased to more than 100% of the initial value, the ER under θ_1_ of 7.04° is greater than other cases. Thus, a larger θ_1_ leads to a greater maximum ER when D5 increases. 

Figure 19 presents the change of ER with XL5 based on optimized D5 under given θ_1_. When XL5 rises from 90% to 95% of its initial size, ER with θ_1_ of 4.62° (60%) is the maximum; however, when the size of XL5 is 100%, the ER with θ_1_ of 7.04° (80%) is the highest; when the size of XL5 increases to 105%, ER with θ_1_ of 4.62° (60%) is still the largest; the ER with θ_1_ of 10.81° (60%) is the maximum when the size of XL5 is increased to 110%. It can be seen that with the varied XL5, there is an optimal θ_1_ to obtain the maximum ER. 

Thus, it can be seen that a larger θ_1_ is needed to achieve a better performance of the ejector when XL3 and D5 increase, however, the relation between ER and θ_1_ is not obvious with the increase of XL5. 

To summarize, with the increase of XL3, a larger θ_1_ and a smaller L_1_ are needed to provide the highest ER; with the increase of D5, it requires larger of them to achieve the maximum ER; however, ER has not an evident relation with varied θ_1_ or L_1_ with given XL5.

### 4.2. Effect of the Diverging Section of the Primary Nozzle

(1)Ejector performance under varied lengths and angles of the diverging section

Similarly, when all else conditions keep unchanged, the impact of the length of the diverging section of the primary nozzle (L_2_) on ER is demonstrated in Figure 20. When L_2_ increases from 20% (2 mm) to 100% (10 mm), ER rises to the maximum of 0.407. Therefore, the optimal length of the diverging section of the primary nozzle is 4 mm, which provides an increase of 31.5% for the ER.

The impact of L_2_ on the ER can be explained by using Mach number contour at varied L_2_ as illustrated in Figure 21, in which, L_2_ is increased from 20% of the initial value to 120% with an interval of 20%. When L_2_ is 20% of the initial value, it can be seen from the figure that there are only four shock clusters, and the highest Mach number of the first shock cluster is 2.82. However, the intensity of the first shock cluster is relatively small and it decays quickly with the increase of shock cluster number, which results in a relatively small entrainment effect of the ejector. When L_2_ increases to 40% of the initial value, although the maximum Mach number in the first shock cluster is only 2.27, the number of shock clusters increases to six, and the intensity of each shock cluster is relatively large. In other words, the attenuation between adjacent shock clusters slows down, and the shock chain length increases significantly. Therefore, the ejector performance reaches the best. However, when L_2_ is increased to 60% of the initial value, although the shock chain length increases slightly, the maximum Mach number decreases to 2.08, and the interaction between two adjacent shock clusters causes a large loss of energy, so the entrainment capacity decreases and the ejector performance decreases accordingly. When L_2_ is increased to 80%, 100%, and 120% of the initial value, the number of shock clusters inside the ejector decreases and the length of the shock chain becomes shorter gradually, and the attenuation between adjacent shock clusters becomes evident, thus, the ejector performance decreases gradually.

Figure 22 illustrates the impact of the angle of a diverging section of the primary nozzle (θ_2_) on ER. As the θ_2_ increases from 20% (0.915°) to 140% (6.404°), ER rises to the maximum of 0.381. Therefore, the optimal θ_2_ is considered to be 1.83° (40% of initial θ_2_), which offers an increase of 22.9% for the ER. 

The impact of θ_2_ on the ER can be explained by using the number and intensity of shock clusters and shock chain length at different θ_2_ as displayed in Figure 23, in which, θ_2_ increases from 20% of the initial value to 120% with an interval of 20%. When θ_2_ is 20% of the initial value, it can be clearly seen that there are four shock clusters. Although the highest Mach number in the first shock cluster reaches 2.87, its overall intensity is small, the attenuation between adjacent shock clusters is very obvious, and the shock chain length is relatively short, so the ejector performance is relatively low. When θ_2_ is increased to 40% of the initial value, the number of shock clusters increases to six. Although the maximum Mach number in the first shock cluster decreases to 2.28, the intensity of the first four shock clusters increases significantly, and the shock chain length increases considerably. Therefore, the ejector performance reaches the best. When θ_2_ increases to 60% of the initial value, the number of shock clusters remains unchanged and the shock chain length increases slightly. However, the maximum Mach number in the first shock cluster decreases to 1.91, and the interference between the two adjacent shock clusters is significantly enhanced, which causes a certain amount of energy loss. Therefore, the suction capacity of the ejector decreases, thus, the ER reduces. Additionally, when θ_2_ continues to increase to 80%, 100%, and 120% of the initial value, although the maximum Mach number gradually increases, the attenuation between adjacent shock clusters increases, the number of shock clusters and the shock chain length gradually decreases, hence, the performance of the ejector gradually decreases.

(2)Individually optimizing each key geometry under varied θ_2_ or L_2_

In this section, XL3, D5, or XL5 is individually optimized under different θ_2_ or L_2_. Figure 24 shows the optimized XL3, D5, and XL5 under different L_2_ when they are individually optimized. According to the results, optimized XL3 decreases, D5 and XL5 increases first and then decreases with L_2_ when L_2_ increases from 20% to 120% of its initial value. 

Figure 25 shows the optimized XL3, D5, and XL5 at different θ_2_, when XL3, D5, and XL5 are separately optimized. It can be observed that optimized XL3, D5, and XL5 increase first and then decrease when θ_2_ increases from 20% (0.92°) to 100% (4.57°) of its initial value. 

(3)Sequentially optimizing three key geometries under varied lengths and angles of nozzle diverging section

In this section, the impact of the angle (θ_2_) and length (L_2_) of the nozzle diverging section on the optimized key dimensions and corresponding optimal ER with the optimization sequence of XL3→D5→XL5 is studied. 

Figure 26 shows two key dimensions optimized with the given optimization sequence under different L_2_. It can be seen that the optimized D5 remains unchanged as L_2_ changes. As L_2_ increases from 0% (0 mm) to 40% (4 mm), the optimal XL5 increases from 90% of the initial size to 115%. However, as L_2_ continues to increase from 40% to 120%, the optimal XL5 decreases linearly from 115% to 95% of its initial value.

Table 4 shows the optimal ER corresponding to the optimized two key dimensions at different L_2_ with the given sequential optimization. It can be seen that with the increase of L_2_, the optimal ER corresponding to optimized D5 is at 4 mm of L_2_. The highest ER is 0.5017 with optimized D5 (100%) when L_2_ is 4 mm (40% of the initial size). In addition, optimal ER corresponding to optimized XL5 has no evident change trend with L_2_, however, the maximum ER is 0.5027 with optimized XL5 (115%) when L_2_ is still 4 mm (40% of the initial size).

Figure 27 shows the percentage of the optimized two key dimensions corresponding to the maximum ER when the two key dimensions are optimized sequentially based on optimized XL3 at different θ_2_. When θ_2_ increases from 0% to 20% of the initial value, the optimized D5 increases from 100% to 101% of the initial value, and when θ_2_ continues to increase 40% (2.74°) of its initial value, the optimized D5 decreases from 101% to 100% of the initial value. When θ_2_ continues to increase to 100% (4.57°), the optimized D5 remains unchanged. Thus, the change of θ_2_ has little impact on the optimized D5. When θ_2_ increases from 0% (0°) to 20% (0.92°), the optimized XL5 decreases from 130% to 125%. When θ_2_ further increases from 20% (0.92°) to 40% (1.83°) of the initial value, the optimized XL5 keeps at 125% of the initial value. With the θ_2_ continuing to increase, the optimized XL5 decreases almost linearly, and finally reaches 100% of the initial XL5 when θ_2_ is 100% of its initial value. In other words, the optimized XL5 nearly decreases with the increase of θ_2_, which is probably caused by the reason that XL5 is optimized based on optimized D5.

Table 5 shows the optimal ER values corresponding to the optimized two key dimensions at different θ_2_. It can be seen from the table that, the optimal ER with each optimized key dimension is obtained with the increase of θ_2_. When θ_2_ is 20% of the initial value (0.92°), ERs reach the maximum values of 0.5098 and 0.5169 when the optimized D5 and XL5 are 101% and 125% of their initial values, respectively. Therefore, the optimal θ_2_ is 20% of the initial value, which can offer the highest ER of 0.5169 when three key dimensions are optimized.

## 5. Conclusions

In this paper, the three key dimensions of XL3, D5, and XL5 are separately and sequentially optimized under different L_1_, θ_1_, L_2_, and θ_2_. Key conclusions are given below:(1)The optimization of L_1_ and θ_1_ can improve the ER by 2.9% and 2.4%, while the optimization of those of the diverging section of the nozzle can lift it by 31.5% and 22.9%, respectively. Thus, the ER improvement is quite limited if only one of the four parameters of the nozzle is optimized.(2)With the increase of XL3, a larger θ_1_ and a smaller L_1_ are needed to offer the maximum ER; with the increase of D5, it requires the larger of them to obtain the highest ER; however, ER has not an evident relation with θ_1_ or L_1_ with given XL5.(3)If XL3, D5, or XL5 is separately optimized based on given θ_2_ or L_2_. Each of the optimized dimensions is identified with a given angle or length of the nozzle diverging section; furthermore, smaller optimized D5 or XL5 can achieve better optimal ER.(4)With the given sequence of XL3→D5→XL5, optimized XL3 and D5 are obtained with given θ_2_; and optimized XL3 and XL5 are achieved with given L_2_. However, two exceptions are different from others, one is that the optimized XL5 decreases almost linearly with the increase of θ_2_; and the other is that the optimized D5 keeps unchanged with L_2_.(5)The ejector ER can be explained well with the number of shock clusters and shock chain length or area of low static pressure zone when the geometries vary.

## Figures and Tables

**Figure 1 entropy-25-00015-f001:**
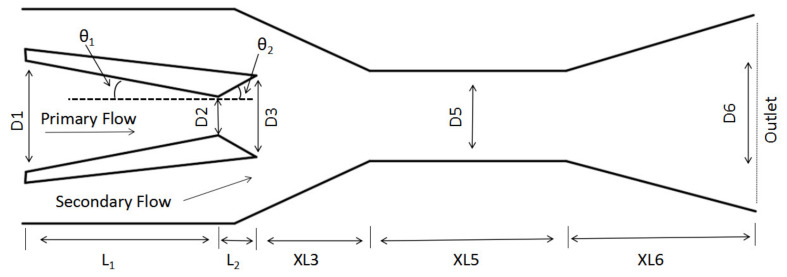
Schematic of the utilized ejector.

**Figure 2 entropy-25-00015-f002:**
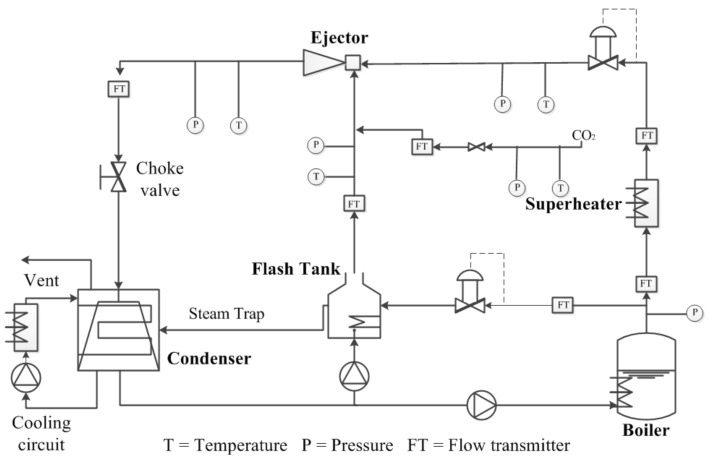
Schematic of the ejector-based carbon capture system.

**Figure 3 entropy-25-00015-f003:**
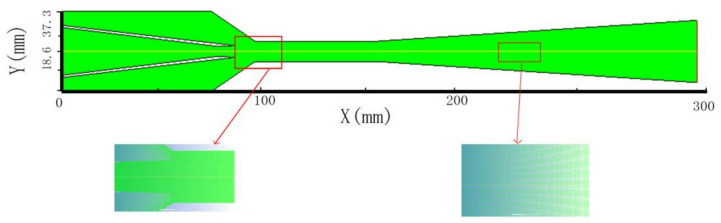
2D axisymmetric quadrilateral grid structure.

**Figure 4 entropy-25-00015-f004:**
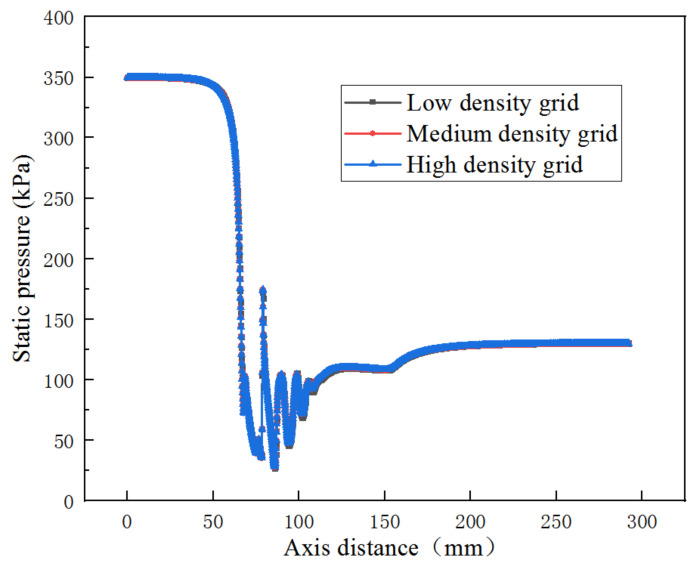
Axial static pressure distribution under low, medium and high grid densities.

**Figure 5 entropy-25-00015-f005:**
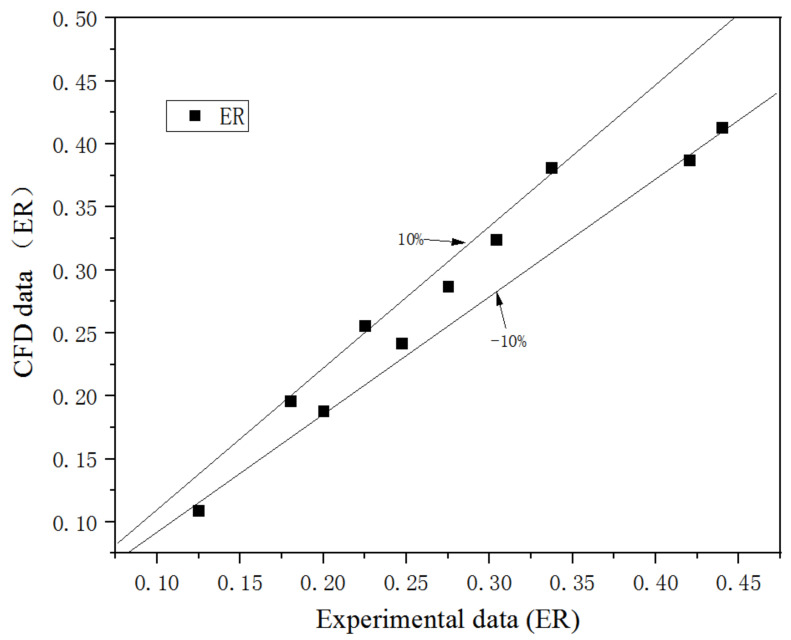
Comparison of CFD and experimental ER.

**Figure 6 entropy-25-00015-f006:**
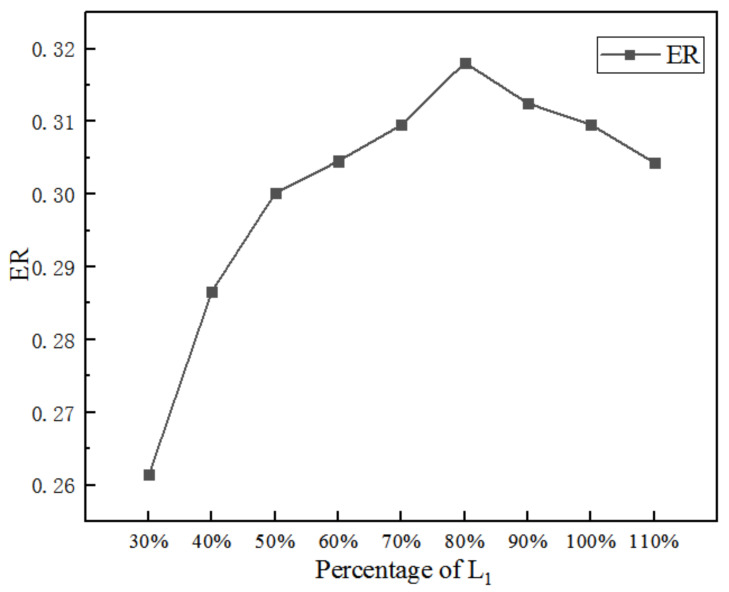
Impact of the L_1_ on ER.

**Figure 7 entropy-25-00015-f007:**
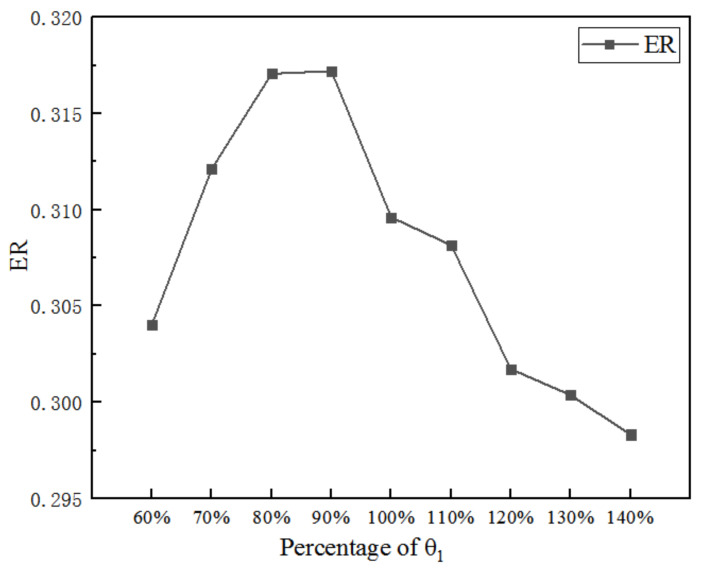
Impact of the θ_1_ on ER.

**Figure 8 entropy-25-00015-f008:**
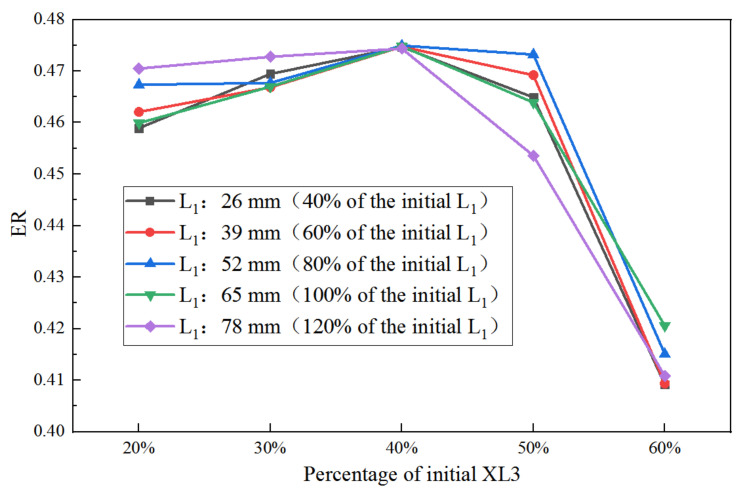
Impact of XL3 on ER as XL3 varies from 20% to 60% of its initial value under varied L_1_.

**Figure 9 entropy-25-00015-f009:**
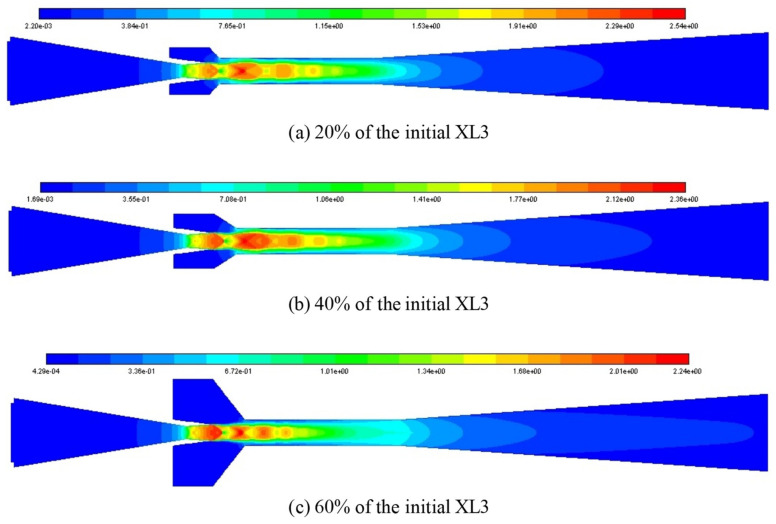
Mach number distribution contour at different XL3 when the L_1_ is 100%.

**Figure 10 entropy-25-00015-f010:**
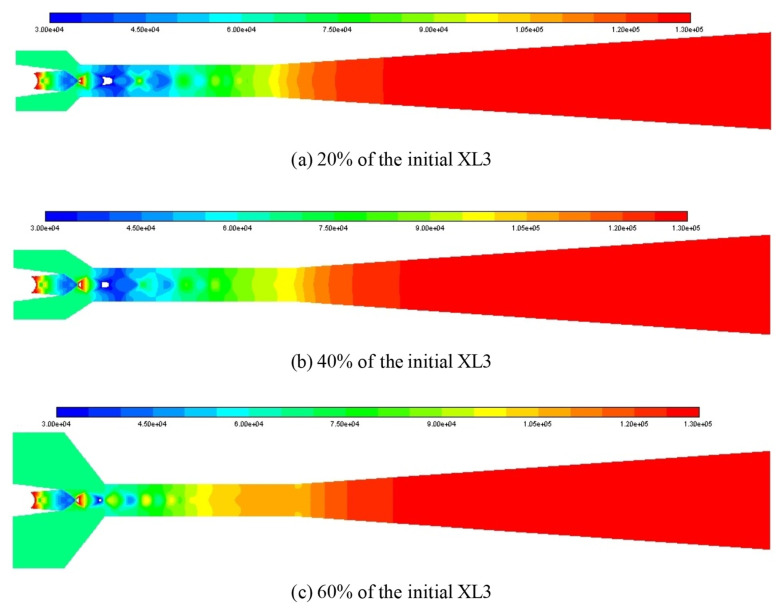
Static pressure (unit: kPa) distribution contour at different XL3 when the L_1_ is 100% of its initial value.

**Figure 11 entropy-25-00015-f011:**
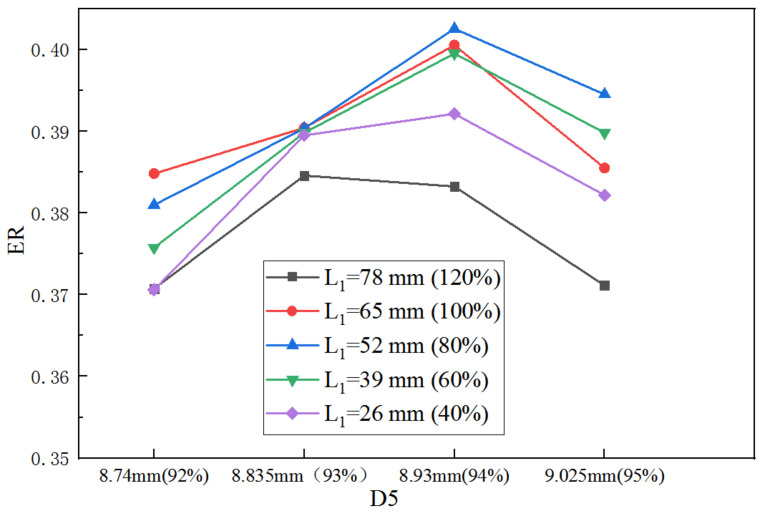
Impact of D5 on ER as D5 varies from 92% to 95% of its initial value under varied L_1_.

**Figure 12 entropy-25-00015-f012:**
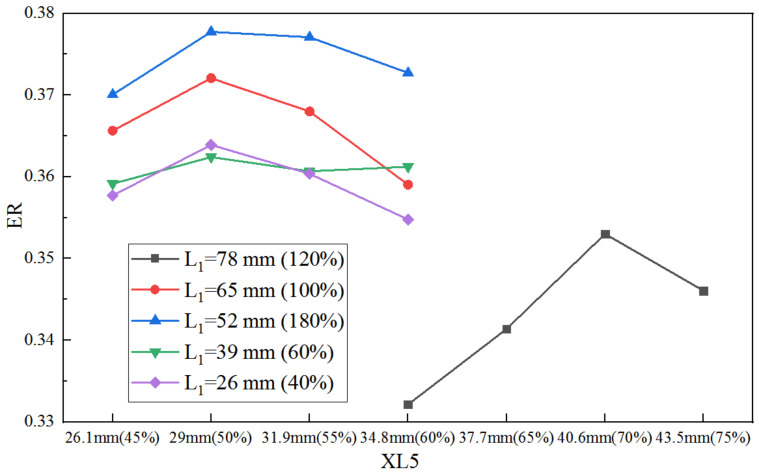
Impact of XL5 on ER as XL5 varies from 45% to 75% of its initial value under varied L_1_.

**Figure 13 entropy-25-00015-f013:**
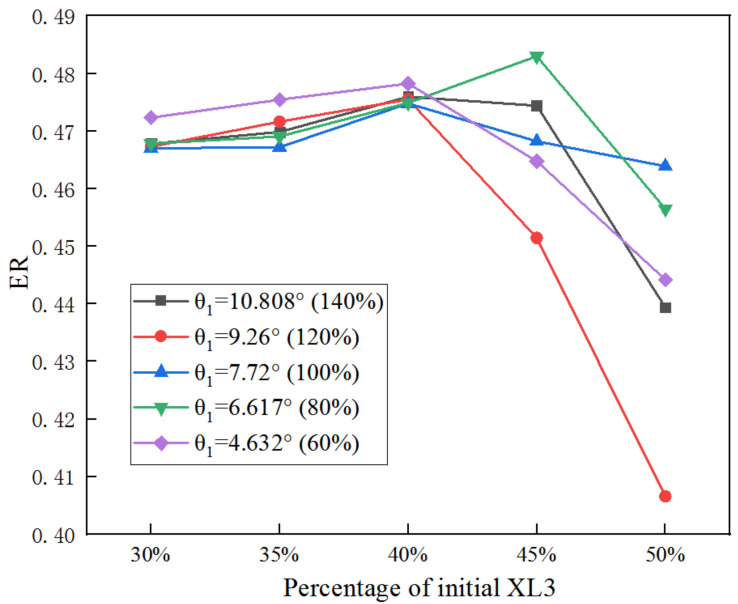
Impact of XL3 on ER as XL3 varies from 30% to 50% of its initial value under varied θ_1_.

**Figure 14 entropy-25-00015-f014:**
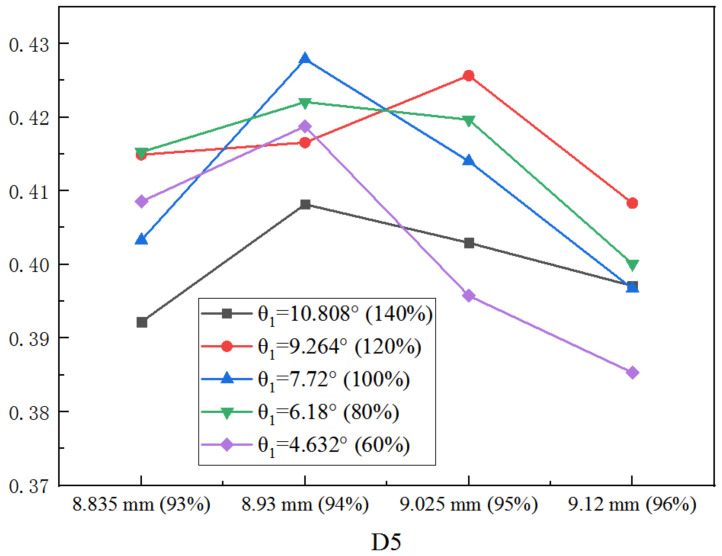
Impact of D5 on ER as D5 varies from 93% to 96% of its initial value under varied θ_1_.

**Figure 15 entropy-25-00015-f015:**
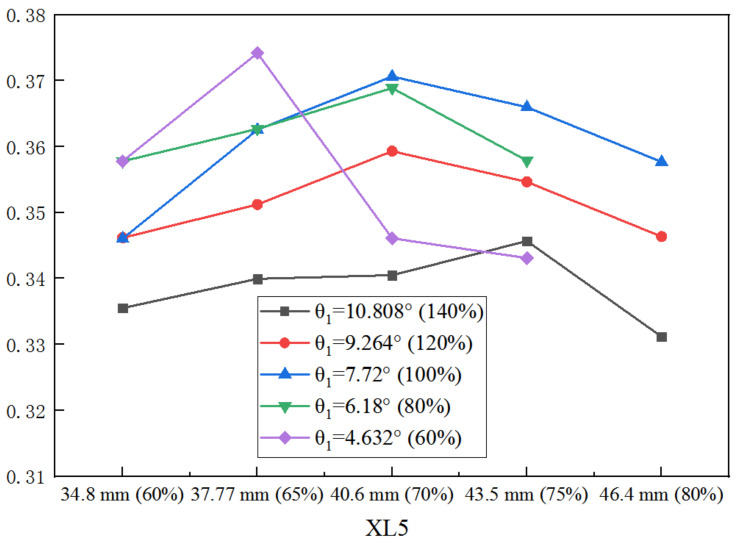
Impact of XL5 on ER as XL5 varies from 60% to 80% of its initial value under varied θ_1_.

**Figure 16 entropy-25-00015-f016:**
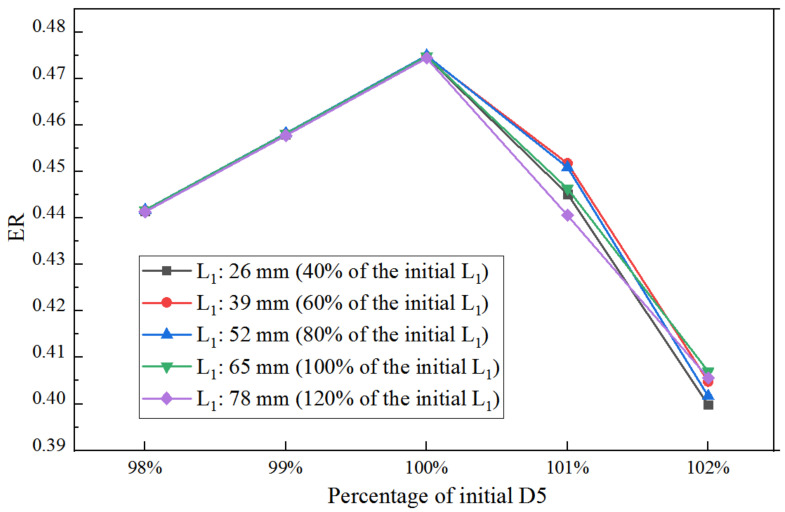
Impact of D5 on ER (with optimized XL3) as L_1_ varies from 40% to 120% of its initial value.

**Figure 17 entropy-25-00015-f017:**
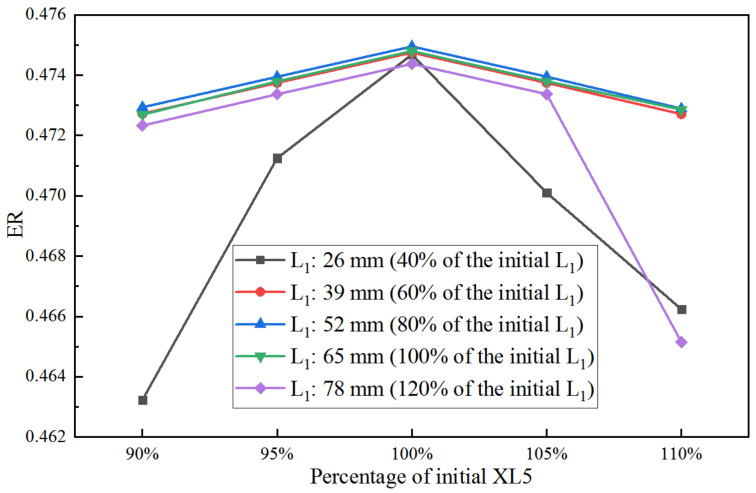
Impact of XL5 on ER (with optimized D5) as L_1_ varies from 40% to 120%.

**Figure 18 entropy-25-00015-f018:**
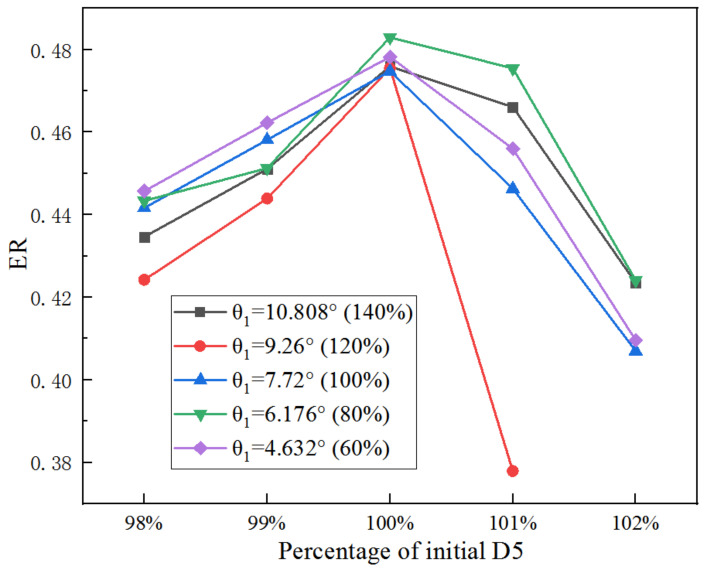
Impact of percentage of D5 on ER (with optimized XL3) when θ_1_ varies from 60% to 140%.

**Figure 19 entropy-25-00015-f019:**
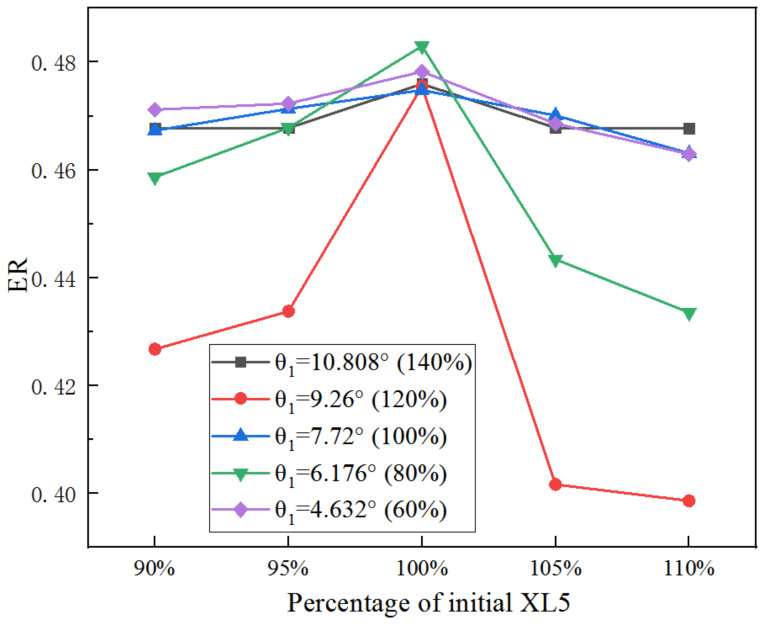
Impact of percentage of XL5 on ER (with optimized D5) as θ_1_ varies from 60% to 140%.

**Figure 20 entropy-25-00015-f020:**
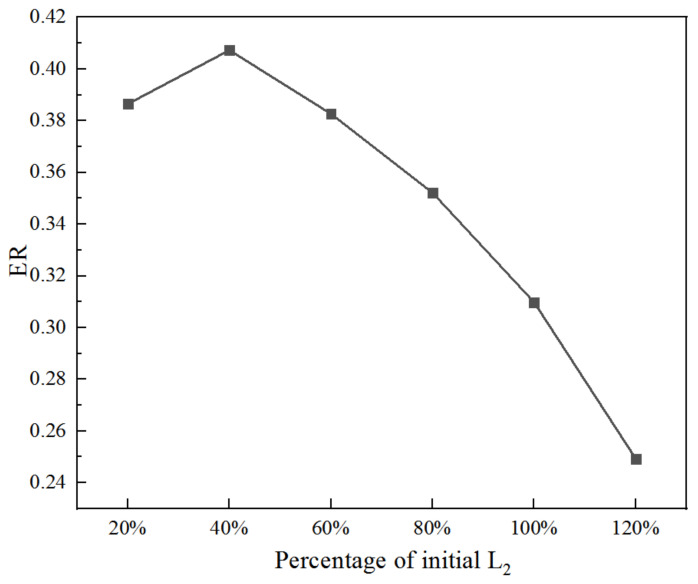
Impact of the length of diverging section of the primary nozzle on ER.

**Figure 21 entropy-25-00015-f021:**
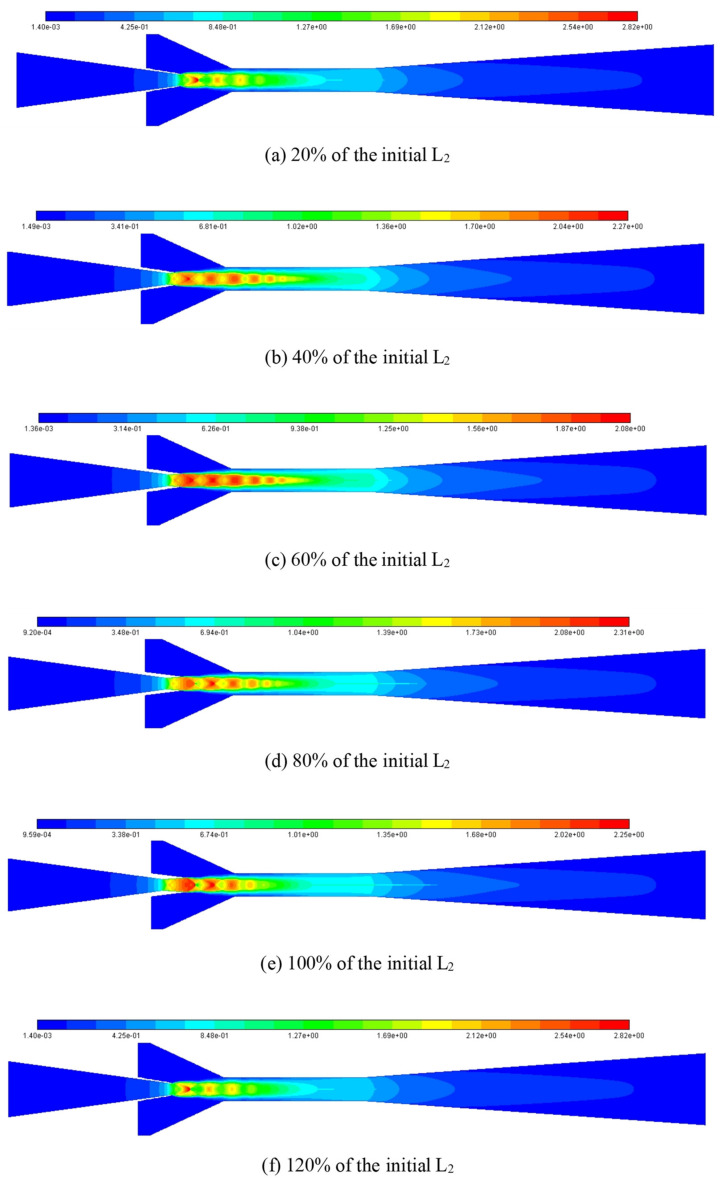
Mach number distribution contour at different L_2_.

**Figure 22 entropy-25-00015-f022:**
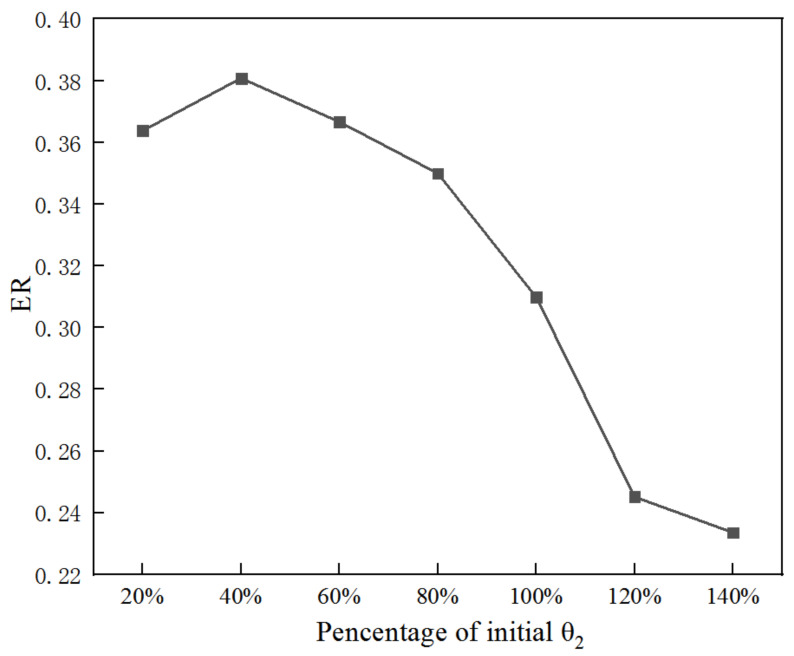
Impact of the angle of diverging section of the primary nozzle on ER.

**Figure 23 entropy-25-00015-f023:**
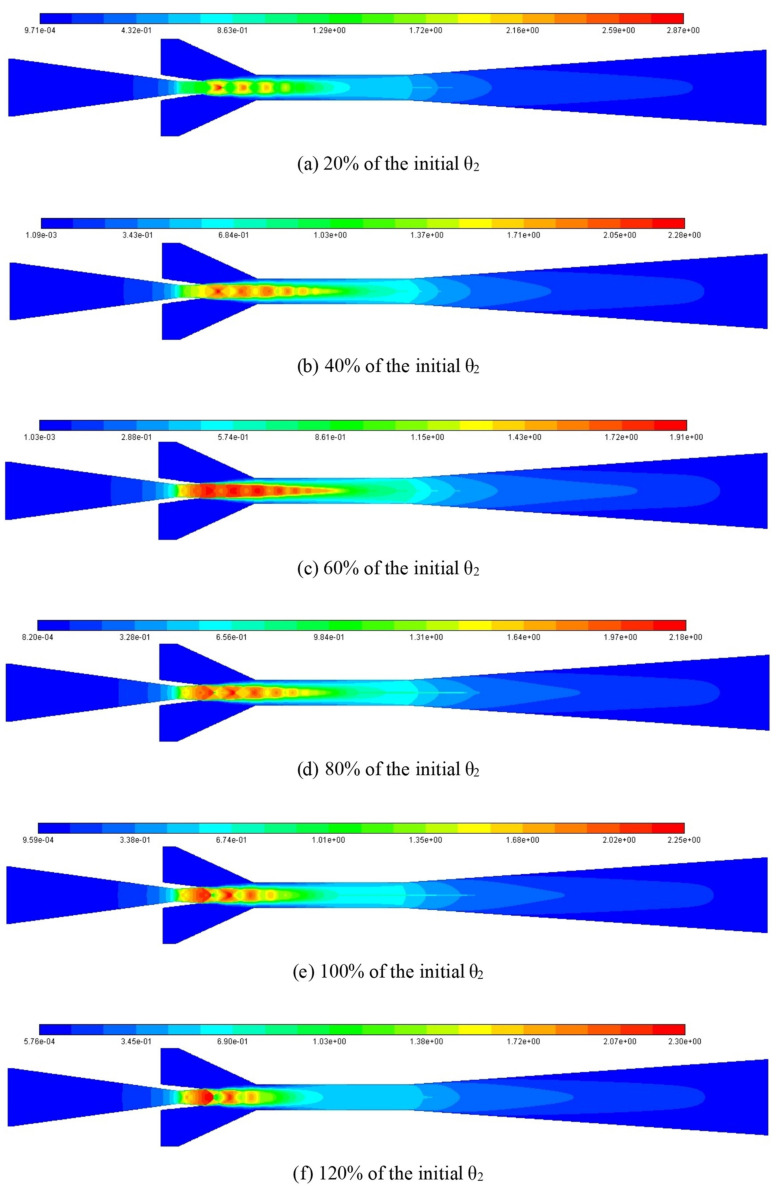
Mach number distribution contour at different θ_2_.

**Figure 24 entropy-25-00015-f024:**
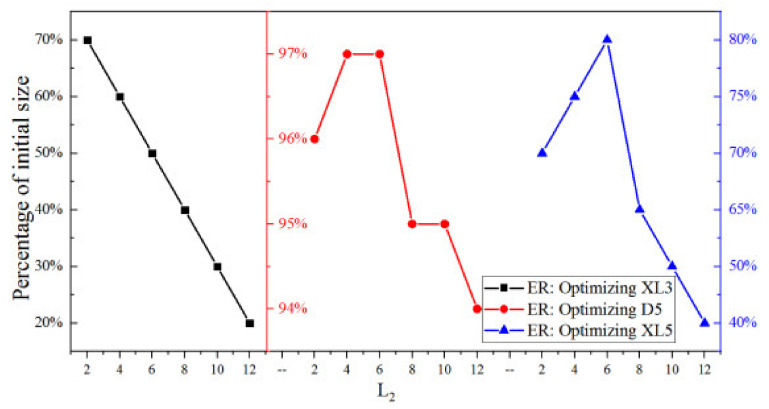
Optimized XL3, D5 and XL5 at different L_2_.

**Figure 25 entropy-25-00015-f025:**
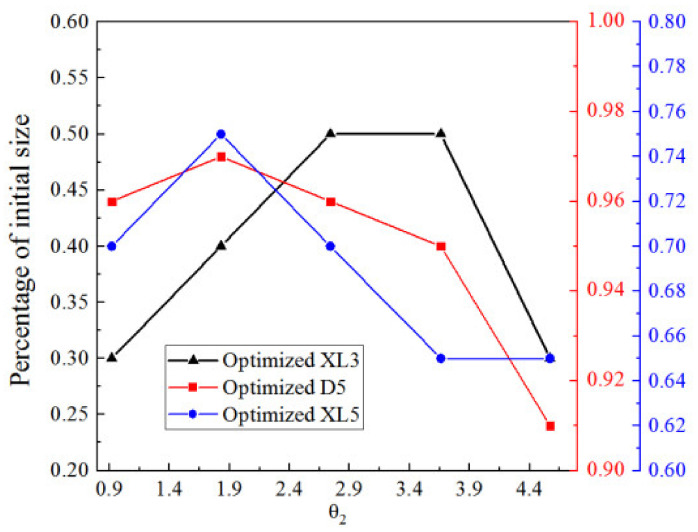
Optimized XL3, D5 and XL5 at different θ_2_.

**Figure 26 entropy-25-00015-f026:**
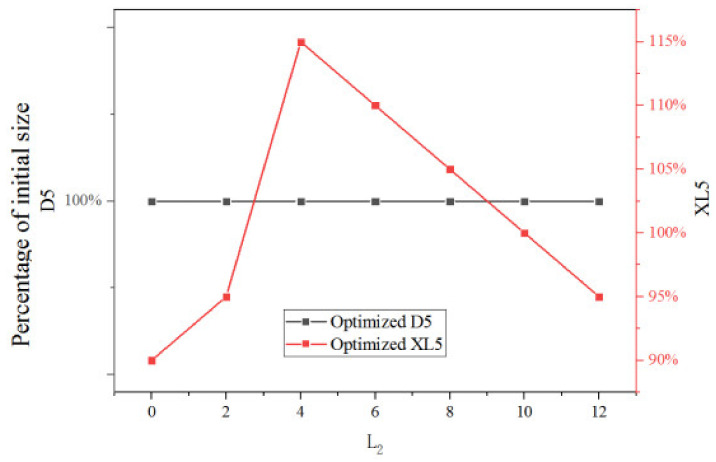
Optimized two key dimensions at different L_2_ (based on optimized XL3).

**Figure 27 entropy-25-00015-f027:**
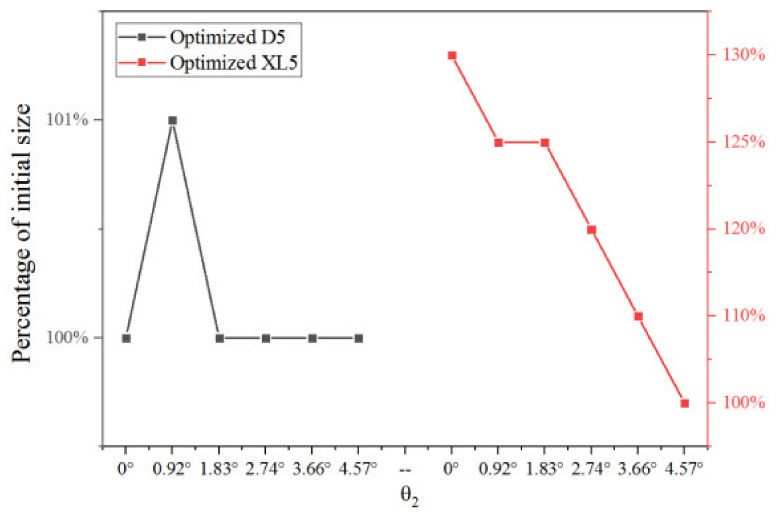
Optimized two key dimensions at different θ_2_ (based on optimized XL3).

**Table 1 entropy-25-00015-t001:** Initial ejector geometries.

Parameters	Value
L_1_	65 mm
θ_1_	7.72°
L_2_	10 mm
θ_2_	4.58°
XL3	20 mm
XL5	58 mm
XL6	139 mm
D1	22.2 mm
D2	6.4 mm
D3	6.2 mm
D5	9.5 mm
D6	29 mm

**Table 2 entropy-25-00015-t002:** Inlet and outlet conditions of the studied ejector (refer to Figure 1).

Primary flow inlet	350 (kPa)	427 (K)
Secondary flow inlet	70 (kPa)	373 (K)
Outlet	130 (kPa)	388 (K)

**Table 3 entropy-25-00015-t003:** ER under different L_1_ when D5 is 98%, 99%, and 100% of the initial value (based on optimized XL3).

Percentage of D5	L_1_				
	26 mm	39 mm	52 mm	65 mm	78 mm
98%	0.44141	0.44160	0.44172	0.44170	0.44135
99%	0.45793	0.45806	0.45821	0.45813	0.45775
100%	0.47468	0.47474	0.47495	0.47480	0.47437

**Table 4 entropy-25-00015-t004:** Optimal ER corresponding to the optimized dimensions under different L_2_ (based on optimized XL3).

Varied L_2_ (mm)	Optimal ER with Optimized D5	Optimal ER with Optimized XL5
0	0.4665	0.4673
2	0.4828	0.4958
4	0.5017	0.5027
6	0.4984	0.4985
8	0.4971	0.5025
10	0.4763	0.4889
12	0.4568	0.4571

**Table 5 entropy-25-00015-t005:** Optimal ER corresponding to the optimized dimensions under different θ_2_ (based on optimized XL3).

Varied θ_2_	Optimal ER with Optimized D5	Optimal ER with Optimized XL5
0°	0.4689	0.4691
0.92°	0.5098	0.5169
1.83°	0.5094	0.5108
2.74°	0.5005	0.5036
3.66°	0.4978	0.4978
4.57°	0.4764	0.4889

## Data Availability

Not applicable.

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
