# Peer review of "Optimization of Three Key Geometries of a Steam Ejector under Varied Primary Nozzle Geometries"

_entropy, 2022, doi:10.3390/e25010015_

Round 1

Reviewer 1 Report

The authors have done some thermodynamics analysis for the steam ejector. I have some concerns about this work

1. Please explain the applications of the steam ejector in the introduction.

2. Improve the state of the art. These papers on numerical study may helpful

https://doi.org/10.1016/j.ijthermalsci.2021.107016

https://doi.org/10.1007/s10973-020-10533-0

3. Add nomenclature section

4. What is the novelty of the study. Explain 

5. What is the y-axis in figure 24 and 25

6. Explain properly, how you optimized the ejector performance

7. So many figures are not required for each case. Please keep the important one

8. Correct it.-------Check the symbol in page 20.

9. what is y axis in Table 7

10.  Please take care of typos and grammatical mistakes.

11. Give a table on one is showing varied variable and the ejector performace. It would be better. 

Reviewer 2 Report

Optimization of Three Key Geometries of a Steam Ejector under Varied Primary Nozzle Geometries

Title:

This title doesn’t reflect the work described in the paper. Can the authors revise it? The meaning of “Primary Nozzle” is not quite clear.

What is “ER”? This was not defined in the abstract.

Introduction:

Page 2:

How exactly does this study defer from your previous studies?

“So far, however, the effect of the primary nozzle structure on the optimum of three key geometries of a steam ejector (Fig. 1) has not been mentioned.”

Before this sentence, it might be useful to describe the ejector depicted in Figure 1, stating the various sections, how they affect the ejector performance and why they have to be optimised.

Experimental Rig:

Page 3:

A more detailed explanation of the experimental rig as depicted in Figure 2, referenced to Figure 1, and how this was carried out would be useful.

CFD modelling and model validation:

Page 4:

It is not enough to refer the reader to your previous publications for details of the CFD modelling used in this paper, please present them and describe them.

Also label your figures appropriately, from Table 2, where can we find the primary, secondary inlets and the exit? No figure in the manuscript shows these. These also feed into the title of the paper which is unclear, as has been pointed out earlier.

Page 5:

If the grid sizes are within 1% of each other, why did you pick the 161,400 cells grid for your simulations over the other grid sizes?

Which particular grid size does Figure 5 refer to?

Results and discussion:

Before commencing discussions regarding your results, it might be useful to define the parameters such ER mathematically, and why they are performance parameters that need to be optimised.

Page 10:

What is the unit for static pressure? Please add this or the figure of figure caption.

These also refer to similar figures throughout the manuscript.

The rest of the manuscript is verbose and difficult to follow. A summary table indicating the various permutations of the parameters studies would be helpful and the authors should then recommend a set of optimal conditions given their observations.

A spelling and grammar check should also be carried out to improve the understanding of the paper.

Round 2

Reviewer 1 Report

References 2 and 3 are the same. 

Please use reference 2:  https://doi.org/10.1007/s10973-020-10533-0 (Keep as it is).

Please use reference 3: https://doi.org/10.1016/j.ijthermalsci.2021.106972

Reviewer 2 Report

Thank you to the authors for revising their manuscript. However, the revisions are insufficient.

The aim of the suggested revisions is to help the authors improve the manuscript.

Optimization of Three Key Geometries of a Steam Ejector under Varied Primary Nozzle Geometries

Part 2

Introduction:

Page 2:

How exactly does this study defer from your previous studies?

You have explained the differences in your cover letter but not on the manuscript. It is not clear how your present paper differs from your previous one; no attempt was made to present this in your manuscript.

“So far, however, the effect of the primary nozzle structure on the optimum of three key geometries of a steam ejector (Fig. 1) has not been mentioned.”

Before this sentence, it might be useful to describe the ejector depicted in Figure 1, stating the various sections, how they affect the ejector performance and why they have to be optimised.

You cannot expect the reader to get this information from your previous paper!

CFD modelling and model validation:

Page 4:

It is not enough to refer the reader to your previous publications for details of the CFD modelling used in this paper, please present them and describe them.

Your description of the CFD modelling details is inadequate! No one can reproduce your CFD model with the information given. Referring the reader to your previous publication is inadequate.

Also label your figures appropriately, from Table 2, where can we find the primary, secondary inlets and the exit? No figure in the manuscript shows these. These also feed into the title of the paper which is unclear, as has been pointed out earlier.

Table 2 should refer to Figure 1 as where the primary, secondary inlets and exit can be found.

Page 5:

If the grid sizes are within 1% of each other, why did you pick the 161,400 cells grid for your simulations over the other grid sizes?

Again, from the manuscript, it is not clear why you picked the 161,400 cells grid. Besides, it is not the medium sized grid; form the figures you presented, it is the largest sized grid.

Which particular grid size does Figure 5 refer to?

On the manuscript, it is clear which grid size was used for Figure 5

Results and discussion:

Before commencing discussions regarding your results, it might be useful to define the parameters such ER mathematically, and why they are performance parameters that need to be optimised.

I am sorry, but this has not been addressed.

Page 10:

What is the unit for static pressure? Please add this or the figure of figure caption.

Please add this unit to your figures!

These also refer to similar figures throughout the manuscript.

The rest of the manuscript is verbose and difficult to follow. A summary table indicating the various permutations of the parameters studies would be helpful and the authors should then recommend a set of optimal conditions given their observations.

The various permutations and parametric studies are difficult to follow as written, therefore a summary of the various permutations should be presented, preferably in a Table format.

A spelling and grammar check should also be carried out to improve the understanding of the paper.

This should still happen
